# Diagnosis of human immunodeficiency virus associated disseminated intravascular coagulation

**Elizabeth S. Mayne**[1,2]*, **Anthony Mayne**[3], **Susan Louw**[2,3]

**1** Division of Immunology, Department of Pathology, Faculty of Health Sciences, University of Cape Town, Cape Town, South Africa, **2** National Health Laboratory Service, Johannesburg, South Africa, **3** Department of Molecular Medicine and Haematology, School of Pathology, Faculty of Health Sciences, University of the Witwatersrand, Johannesburg, South Africa

* elizabeth.mayne@nhls.ac.za

**Data Availability Statement:** All data are summarised in this paper.

**Funding:** ESM Discovery Academic incidence award South African National Research Foundation Thuthuka award -TTK20110801000022866 The

## Abstract

### Introduction

Disseminated intravascular Coagulation (DIC) is a thrombotic microangiopathy which may complicate a number of severe disease processes including sepsis. Development of microvascular thromboses results in consumption of coagulation factors and platelets and ultimate bleeding. Patients with HIV infection (PWH) often present with baseline dysregulation of the coagulation system which may increase severity and derangement of DIC presentation. Previously, we have shown that HIV is a significant risk factor for development of DIC.

### Methodology

We conducted a retrospective record review of all DIC screens submitted to our tertiary coagulation laboratory in Johannesburg, South Africa, over a one year period and compared the laboratory presentation of DIC in PWH with presentation of DIC in patients without HIV infection.

### Results

Over the year, 246 patients fulfilled the International Society of Thrombosis and Haemostasis (ISTH) diagnostic criteria for DIC– 108 were confirmed HIV-infected and 77 were confirmed uninfected. PWH and DIC presented at a significantly earlier age (41 vs 46 years respectively, p<0.02). The prothrombin time was significantly more prolonged (30.1s vs 26.s), the d-dimer levels were substantially higher (5.89mg/L vs 4.52mg/L) and the fibrinogen (3.92g/L vs 1.73g/L) and platelet levels (64.8 vs 114.8x10⁹/l) were significantly lower in PWH. PWH also showed significant synthetic liver dysfunction and higher background inflammation.

funders had no role in study design, data collection and analysis, decision to publish or preparation of the manuscript.

**Competing interests:** The authors have declared that no competing interests exist.

## Conclusion

PWH who fulfil the diagnostic criteria for DIC show significantly more dysregulation of the haemostatic system. This may reflect baseline abnormalities including endothelial dysfunction in the context of inflammation and liver dysfunction.

## Introduction

South Africa has a high prevalence of Human immunodeficiency virus (HIV) infections with an estimated 13% or 7.8 million people with HIV (PWH) and approximately 110 000 HIV-related deaths annually [1]. The prevalence of HIV infection is even higher in the South African in-hospital population particularly in medical wards and intensive care units where the true prevalence may be over 60% [2, 3]. HIV infection status should be considered in all patients particularly since HIV can result in unusual presentations and exacerbations of pre-existing conditions [4].

HIV infection is strongly associated with pro-thrombotic states including pulmonary thromboembolic disease and deep vein thrombosis [5], arterial thrombosis manifesting as cardiovascular disease [6] and microvascular disorders including, most prominently, thrombotic thrombocytopaenic purpura (TTP) [7]. This increased propensity to pathological clotting has been attributed to a number of different features including endothelial dysfunction resulting from chronic inflammation (reviewed in [8]), associated infections (opportunistic and non-opportunistic), an imbalance between pro- and anti-coagulant factor levels [9–15] as well as platelet dysfunction [16]. These derangements, documented in individuals on antiretroviral therapy (ART) as well as in patients with uncontrolled viraemia (ART-naïve), include decreased Protein S levels, elevated levels of coagulation factors including factor VIII and von Willebrand factor [9, 14] and quantitative and qualitative platelet disorders [13, 16, 17]. The imbalance between pro- and anticoagulant factors manifests in the laboratory as elevated D-dimers levels in HIV infected individuals [18, 19].

Disseminated intravascular coagulation (DIC) is a thrombotic microangiopathic state characterised by widespread microvascular thrombosis, consumption of coagulation factors and platelets and ultimately a bleeding diathesis [20]. It is important to make the diagnosis of DIC in an appropriate clinical setting and a number of triggers have been associated with DIC development including severe infection and sepsis, malignancy, trauma and obstetric complications [21]. No single laboratory test is sufficiently robust, specific or sensitive enough to diagnose DIC. The diagnosis may be made using a number of scoring systems which assign numeric values to abnormalities in a panel of tests including the International Society of Thrombosis and Haemostasis (ISTH) Diagnostic Scoring system which assigns points for reduction in platelet count, elevation of fibrin degradation products, prolongation of the prothrombin time and the fibrinogen level (Table 1) [20].

Although HIV is associated with the development of DIC, this has been considered an indirect relationship with the majority of HIV-associated DIC cases attributed to HIV- related infections or malignancies rather than with HIV as a primary pathophysiological trigger for DIC development. We have reported the high risk of development of DIC in HIV-infected patients even in the absence of other comorbidities [21]. It is important to define the trigger accurately as the primary therapy of DIC is the treatment of the underlying cause although supportive replacement of coagulation factors and platelets may also be considered especially

**Table 1. The ISTH scoring system for overt DIC.**

| Parameter and value | Points allocation |
|---|---|
| **Platelet Count** | |
| • >100 x $10^9$/L | • 0 points |
| • <100 x $10^9$/L | • 1 point |
| • <50 x $10^9$/L | • 2 points |
| **Elevation of Fibrin Degradation products (D-dimer levels)** | |
| • No increase | • 0 points |
| • Moderate increase | • 2 points |
| • Strong (marked) increase | • 3 points |
| **Prolongation of the Prothrombin Time (PT)** | |
| • <3 seconds | • 0 points |
| • >3 but <6 seconds | • 1 point |
| • >6 seconds | • 2 points |
| **Fibrinogen level** | |
| • >1 g/L | • 0 points |
| • <1 g/L | • 1 point |
| Score: | |
| ≥ 5: compatible with overt DIC: repeat score daily | |
| <5: suggestive for non-overt DIC: repeat score next 1–2 days | |

if a patient is actively bleeding [22]. PWH often show baseline activation of the coagulation and haemostatic systems which may impact the laboratory presentation of these patients.

In order to define the laboratory presentation of DIC in patients with and without HIV infection, we undertook a retrospective analysis of all DIC screens submitted to our tertiary care academic facility in Johannesburg, South Africa.

## Methodology

Permission for this retrospective study was granted by the University of the Witwatersrand Human Research Ethics Committee (Certificate number: M160389). All data were retrieved from the laboratory information system for screens submitted to the coagulation laboratory at the National Health Laboratory Service (NHLS) Charlotte Maxeke Johannesburg Academic Hospital (CMJAH) for a one year period from 2015 to 2016. This laboratory is a specialist referral facility for the academic hospital and a number of surrounding hospitals in the greater Gauteng area.

Since the study was a retrospective record review, no patient clinical data were obtained and no further testing was possible. All data were anonymized. The minimum dataset used included D-dimer levels, prothrombin time (PT), activated partial thromboplastin time (aPTT), platelet count and fibrinogen levels. The coagulation analysis was performed on a STAGO STA-R Max™ (Diagnostica Stago, Asnières-sur-Seine, France) and platelet counts were performed on the Sysmex XN analyser (Sysmex, Kobe, Japan). For each participant, the ISTH DIC scoring system was used to assess the presence or absence of DIC with patients with ISTH scores of 5 and greater considered to have overt DIC. Other data collected from the database included albumin levels (a surrogate of liver dysfunction), C-reactive protein levels, HIV status (where this had been performed) and CD4+ T cell counts and viral load levels in patients who were HIV infected. Screens where no testing for HIV had been conducted were excluded from the final analysis.

**Table 2. Demographic and laboratory parameters in patients presenting with a DIC.**

| | HIV-infected patients (n = 108) | HIV-uninfected patients (n = 77) | P-value* | Adjusted p-value |
|---|---|---|---|---|
| Age at diagnosis (mean± SD) | 41± 11 | 47± 18 | <0.02 | <0.001 |
| Sex (female; male) | 59; 49 | 44;33 | <0.15 | 0.42 |
| ISTH score (mean± SD) | 5.97± 0.89 | 5.74± 0.85 | >0.50 | 1.53 |
| CD4+ T cells per mm$^3$ (mean± SD)** | 159± 285 | - | - | - |
| Viral Load, log copies/mL (mean±SD)** | 685,375± 1.4x10$^6$ | - | - | - |
| **ISTH DIC score parameters** | | | | |
| PT (s, mean± SD) | 30.12± 34.26 | 27.30±27.45 | <0.05 | 0.28 |
| D-dimer (mg/L, mean± SD) | 5.89± 10.89 | 4.52± 5.77 | <0.04 | 0.22 |
| Fibrinogen (g/L, mean± SD) | 3.92± 1.73 | 4.70± 2.64 | <0.01 | <0.001 |
| Platelets (x10$^9$/L, mean±; SD) | 64.87±91.15 | 114.84± 218.36 | <0.01 | 0.01 |
| **Additional laboratory parameters** | | | | |
| C-reactive protein (mg/L, mean± SD) | 116.98± 83.05 | 115.38± 87.87 | <0.12 | 0.69 |
| Albumin (g/L, mean± SD) | 26.75± 7.89 | 28.00± 10.39 | <0.01 | 0.01 |

N–number, SD–standard deviation, ISTH–International Society of Thrombosis and Haemostasis, PT–prothrombin time

* p-value of <0.05 considered significant

** CD4+ T cell count and viral load available on 96 patients and 67 patients respectively

Statistical analysis was performed using StataSE® 14.2 StataCorp. Summary statistics of all analytes including mean values and standard deviations were computed. Mean values were compared using a student's t-test after applying ranking by an upward sort to analytes by positive or negative HIV status. A *p*-value <0.05 was considered significant.

## Results

For the period, 246 patients met the ISTH diagnostic criteria for a DIC. Of these, 61 had no recorded HIV test and were excluded from further analysis. 108 patients were HIV-infected and 77 patients were confirmed HIV-uninfected. Of the 108 HIV-infected patients, 67 viral loads and 96 CD4+ T cell counts were available. It was not possible to assess treatment status or length of infection in these patients.

The summary data are presented in Table 2. Patients with HIV-associated DIC presented at a significantly earlier age. The coagulation parameters were significantly more deranged although this did not impact the mean ISTH score. Both HIV-infected and uninfected patients presented with a high C-reactive protein, reflecting the high level of inflammation in both cohorts. Importantly, HIV-infected patients had significantly lower mean albumin levels.

## Discussion

HIV-infected patients in South Africa often present for treatment with advanced disease at lower CD4+ T cell counts and with higher viral loads [23, 24]. This is associated both with an increased incidence of opportunistic and non-communicable complications of HIV infection and with longer and more severe chronic inflammation. This results in an underlying derangement of the coagulation pathways. Previously we have shown that HIV-infection can be a significant trigger for the development of the thrombotic microangiopathy (TMA), DIC [21]. In this study, we show that HIV-infected patients with DIC present at a younger age and with a more significant coagulation disorder. Although this did not significantly impact their DIC score, it is possible that the diagnosis of overt DIC may be made erroneously in these patients with significant baseline activation of the haemostatic systems. The implications of the severity

of the coagulopathy on outcomes and management of these patients should be urgently investigated.

Importantly, HIV-infected patients showed significantly lower mean albumin levels. Albumin is a negative acute phase reactant and is often reduced in severe infection. It may also, however, reflect liver synthetic dysfunction [25]. The majority of coagulation factors are synthesised in the liver and the prothrombin time, in particular, is a sensitive indicator of liver disease. Unfortunately, limited clinical data were available but liver involvement in HIV could be associated with opportunistic infections especially disseminated mycobacterial disease or malignancies like, for example, B-cell lymphoma [26–28]. As these conditions may also predispose patients to the development of DIC, this may contribute to diagnostic uncertainty in patients with HIV-associated DIC.

This study has a number of limitations. There were limited clinical data and the outcomes of the patients included is unknown. In the majority of cases, serial measurements were not available and HIV viral loads and CD4+ T cell counts were also not available on all patients. This study does, however, indicate that DIC in PWH presents with more significant derangement of coagulation parameters and this should be considered when making the diagnosis of DIC in this population.

## Supporting information

**S1 File. Raw data for all quantitative DIC parameters.**
(XLSX)

## Author Contributions

**Conceptualization:** Elizabeth S. Mayne, Susan Louw.

**Data curation:** Elizabeth S. Mayne.

**Formal analysis:** Elizabeth S. Mayne, Anthony Mayne, Susan Louw.

**Funding acquisition:** Elizabeth S. Mayne.

**Methodology:** Elizabeth S. Mayne, Susan Louw.

**Writing – original draft:** Elizabeth S. Mayne, Anthony Mayne, Susan Louw.

**Writing – review & editing:** Elizabeth S. Mayne, Anthony Mayne, Susan Louw.

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
