## [Decision Letter · Decision Letter 0]

13 Dec 2021

PONE-D-21-28106Diagnosis of Human Immunodeficiency Virus associated Disseminated Intravascular CoagulationPLOS ONE

Dear Dr. Mayne,

Thank you for submitting your manuscript to PLOS ONE. After careful consideration, we feel that it has merit but does not fully meet PLOS ONE’s publication criteria as it currently stands. Therefore, we invite you to submit a revised version of the manuscript that addresses the points raised during the review process.

We look forward to receiving your revised manuscript.

Kind regards,

Eliseo A Eugenin, Ph.D.

Academic Editor

PLOS ONE

Journal Requirements:

Additional Editor Comments:

HI

Please add a comment or data about the requested information

Best Regards

Eliseo Eugenin

Reviewers' comments:

Reviewer's Responses to Questions

**Comments to the Author**

1. Is the manuscript technically sound, and do the data support the conclusions?

Reviewer #1: Yes

Reviewer #2: Yes

2. Has the statistical analysis been performed appropriately and rigorously? 

Reviewer #1: Yes

Reviewer #2: Yes

3. Have the authors made all data underlying the findings in their manuscript fully available?

Reviewer #1: Yes

Reviewer #2: Yes

4. Is the manuscript presented in an intelligible fashion and written in standard English?

Reviewer #1: Yes

Reviewer #2: Yes

5. Review Comments to the Author

Reviewer #1: The manuscript is brief but well written. However, the authors could have expanded more on the following points:

Do the authors know when each of the 77 HIV infected participants was infected? This could affect the parameters under investigation.

How many of the HIV+ participants were on treatment and for how long?

Age was significantly different between groups, can the authors also present adjusted p-values too?

Reviewer #2: The manuscript from Mayne and colleagues is undoubtedly interesting, since addresses the impact of HIV infection on the severity of DIC presentation.

Data are solid, methodology and scoring system for DIC manifestation is clear.

However, in my opinion paper needs some corrections.

Abstract: section “Results” there is no a period at the end of a sentence.

Introduction and Discussion: Please correct the references in the text, very often is .[1], should be [1]. Choose one style.

Also there are missed spaces between words and periods at the end of a sentences.

Second paragraph of “Introduction”. Sentence – The imbalance between pro- and anticoagulant factors manifests in the laboratory as elevated D-dimers levels in HIV infected individuals. Please add the citation.

Table 2. Is difficult to read. Please, considered to change the display of results from “mean; SD” to “mean±SD”.

Also, there is an information about number of patients samples used for a CD4+ T-cells count and viral load. Is confusing, especially when is called “count”, please change it or remove from the table, there is an information in the text.

Discussion: There is no explanation for TMA, also I couldn't found it in a previous paper Mayne, 2018 [19]

6. PLOS authors have the option to publish the peer review history of their article (what does this mean?). If published, this will include your full peer review and any attached files.

Reviewer #1: No

Reviewer #2: No

---

## [Author Response · Author response to Decision Letter 0]

17 Dec 2021

Response to Reviewers (reviewer comments in bold and responses underlined)

Reviewer #1: The manuscript is brief but well written. However, the authors could have expanded more on the following points:

Do the authors know when each of the 77 HIV infected participants was infected? This could affect the parameters under investigation. How many of the HIV+ participants were on treatment and for how long?

As this was a retrospective study, it was difficult to gauge clinical details which were not available on the laboratory information system. This has been described as a limitation, as below

This study has a number of limitations. There were limited clinical data and the outcomes of the patients included is unknown. In the majority of cases, serial measurements were not available and HIV viral loads and CD4+ T cell counts were also not available on all patients. This study does, however, indicate that DIC in PWH presents with more significant derangement of coagulation parameters and this should be considered when making the diagnosis of DIC in this population. 

In order to emphasise the point, the following sentence has been added to the results section

For the period, 246 patients met the ISTH diagnostic criteria for a DIC. Of these, 61 had no recorded HIV test and were excluded from further analysis. 108 patients were HIV-infected and 77 patients were confirmed HIV-uninfected. Of the 108 HIV-infected patients, 67 viral loads and 96 CD4+ T cell counts were available. It was not possible to assess treatment status or length of infection in these patients. 

Age was significantly different between groups, can the authors also present adjusted p-values too?

A Bonferroni calculation was performed and the adjusted p values are included in Table 2 as below:

 HIV-infected patients (n=108) HIV-uninfected patients (n=77) P-value* Adjusted p-value

Age at diagnosis (mean± SD) 41± 11 47± 18 <0.02 <0.001

Sex (female; male) 59± 49 44± 33 <0.15 0.42 

ISTH score (mean± SD) 5.97± 0.89 5.74± 0.85 >0.50 1.53 

CD4+ T cells per mm3 

(mean± SD)** 

159± 285 

 - 

 - 

-

Viral Load, log copies/mL ( mean±SD)** 

 685,375± 1.4x106 

 - 

 - 

-

ISTH DIC score parameters 

PT (s, mean± SD) 30.12± 34.26 27.30±27.45 <0.05 0.28

D-dimer (mg/L, mean± SD) 5.89± 10.89 4.52± 5.77 <0.04 0.22

Fibrinogen (g/L, mean± SD) 3.92± 1.73 4.70± 2.64 <0.01 <0.001

Platelets (x109/L, mean±; SD) 64.87±91.15 114.84± 218.36 <0.01 0.01

Additional laboratory parameters 

C-reactive protein (mg/L, mean± SD) 116.98± 83.05 115.38± 87.87 <0.12 0.69

Albumin (g/L, mean± SD) 26.75± 62.29 28.00± 108.00 <0.01 0.01

Reviewer #2: The manuscript from Mayne and colleagues is undoubtedly interesting, since addresses the impact of HIV infection on the severity of DIC presentation.

Data are solid, methodology and scoring system for DIC manifestation is clear.

However, in my opinion paper needs some corrections.

Abstract: 

section “Results” there is no a period at the end of a sentence.

Period has been added as follows:

PWH also showed significant synthetic liver dysfunction and higher background inflammation.

Introduction and Discussion: Please correct the references in the text, very often is .[1], should be [1]. Choose one style.

This has been corrected throughout

Also there are missed spaces between words and periods at the end of a sentences.

This has been corrected throughout

Second paragraph of “Introduction”. Sentence – The imbalance between pro- and anticoagulant factors manifests in the laboratory as elevated D-dimers levels in HIV infected individuals. Please add the citation.

Two citations have been added to this sentence namely:

18. Aranda F, Peres Wingeyer S, de Larranaga G. D-Dimer as a prognostic marker of morbidity and mortality among HIV patients: a call for attention. Infect Dis (Lond). 2016;48(11-12):860-1.

19. Borges AH, O'Connor JL, Phillips AN, Baker JV, Vjecha MJ, Losso MH, et al. Factors associated with D-dimer levels in HIV-infected individuals. PLoS One. 2014;9(3):e90978.

Table 2. Is difficult to read. Please, considered to change the display of results from “mean; SD” to “mean±SD”.

Also, there is an information about number of patients samples used for a CD4+ T-cells count and viral load. Is confusing, especially when is called “count”, please change it or remove from the table, there is an information in the text.

Mean; SD has been replaced with Mean±SD throughout

Count has been removed from table and added as an explanatory footnote as below:

Table 2 Demographic and laboratory parameters in patients presenting with a DIC

 HIV-infected patients (n=108) HIV-uninfected patients (n=77) P-value*

Age at diagnosis (mean± SD) 41± 11 47± 18 <0.02

Sex (female; male) 59± 49 44± 33 <0.15

ISTH score (mean± SD) 5.97± 0.89 5.74± 0.85 >0.50

CD4+ T cells per mm3 

(mean± SD)** 

159± 285 

 - 

 -

Viral Load, log copies/mL ( mean±SD)** 

 685,375± 1.4x106 

 - 

 -

ISTH DIC score parameters 

PT (s, mean; SD) 30.12± 34.26 27.30±27.45 <0.05

D-dimer (mg/L, mean± SD) 5.89± 10.89 4.52± 5.77 <0.04

Fibrinogen (g/L, mean± SD) 3.92± 1.73 4.70± 2.64 <0.01

Platelets (x109/L, mean±; SD) 64.87±91.15 114.84± 218.36 <0.01

Additional laboratory parameters

C-reactive protein (mg/L, mean± SD) 116.98± 83.05 115.38± 87.87 <0.12

Albumin (g/L, mean± SD) 26.75± 62.29 28.00± 108.00 <0.01

N – number, SD – standard deviation, ISTH – International Society of Thrombosis and Haemostasis, PT – prothrombin time

* p-value of <0.05 considered significant 

** CD4+ T cell count and viral load available on 96 patients and 67 patients respectively

Discussion: There is no explanation for TMA, also I couldn't found it in a previous paper Mayne, 2018 [19]

This abbreviation has been defined (as below)

Previously we have shown that HIV-infection can be a significant trigger for the development of the thrombotic microangiopathy (TMA), DIC [19].

---

## [Editor Report · Decision Letter 1]

21 Dec 2021

Diagnosis of Human Immunodeficiency Virus associated Disseminated Intravascular Coagulation

PONE-D-21-28106R1

Dear Dr. Mayne,

We’re pleased to inform you that your manuscript has been judged scientifically suitable for publication and will be formally accepted for publication once it meets all outstanding technical requirements.

Kind regards,

Eliseo A Eugenin, Ph.D.

Academic Editor

PLOS ONE

Additional Editor Comments (optional):

Dear Dr. Mayne

Thank you for the comments and suggestions

Eliseo Eugenin

---

## [Editor Report · Acceptance letter]

13 Jan 2022

PONE-D-21-28106R1 

Diagnosis of Human Immunodeficiency Virus associated Disseminated Intravascular Coagulation 

Dear Dr. Mayne:

I'm pleased to inform you that your manuscript has been deemed suitable for publication in PLOS ONE. Congratulations! Your manuscript is now with our production department. 

Kind regards, 

on behalf of

Dr. Eliseo A Eugenin 

Academic Editor

PLOS ONE